# Improving SWAT Model Calibration Using Soil MERGE (SMERGE)

**Kenneth J. Tobin \* and Marvin E. Bennett**

Center for Earth and Environmental Studies, Texas A&M International University, Laredo, TX 78045, USA;
mbennett@tamiu.edu
\* Correspondence: ktobin@tamiu.edu; Tel.: +1-956-326-2417

**Abstract:** This study examined eight Great Plains moderate-sized (832 to 4892 km$^2$) watersheds. The Soil and Water Assessment Tool (SWAT) autocalibration routine SUFI-2 was executed using twenty-three model parameters, from 1995 to 2015 in each basin, to identify highly sensitive parameters (HSP). The model was then run on a year-by-year basis, generating optimal parameter values for each year (1995 to 2015). HSP were correlated against annual precipitation (Parameter-elevation Regressions on Independent Slopes Model—PRISM) and root zone soil moisture (Soil MERGE—SMERGE 2.0) anomaly data. HSP with robust correlation (r > 0.5) were used to calibrate the model on an annual basis (2016 to 2018). Results were compared against a baseline simulation, in which optimal parameters were obtained by running the model for the entire period (1992 to 2015). This approach improved performance for annual simulations generated from 2016 to 2018. SMERGE 2.0 produced more robust results compared with the PRISM product. The main virtue of this approach is that it constrains parameter space, minimizesing equifinality and promotesing modeling based on more physically realistic parameter values.

**Keywords:** SMERGE 2.0; PRISM; root zone soil moisture; SWAT; US Great Plains; mass balance

## 1. Introduction

The Soil and Water Assessment Tool (SWAT) is a physically based model with demonstrated global applications and has been validated at the watershed scale through the publication of thousands of referred papers (see [1]). The SWAT model is moderate in terms of complexity, i.e., it is a semi-distributed model where the watershed is divided into subbasins, in which water balance is calculated on a daily basis. Many SWAT modeling studies have focused on matching simulated and observed streamflow at the basin's outlet. Calibration based on multiple gauges within a basin has been demonstrated to more realistically capture surface flow throughout an entire watershed (e.g., [2,3]). However, this approach, while an improvement, can fail to provide a realistic depiction of landscape conditions that strongly influence runoff production. During recent years, hydrologists have begun to leverage remote sensing observations to improve model calibration and achieve a more accurate picture of processes at a watershed scale. Examples of such studies that utilized the SWAT model span diverse aspects of the hydrologic cycle and include quantifying total terrestrial water [4,5], soil moisture [6–8], evapotranspiration [9–11], and groundwater recharge [12,13].

Since SWAT was designed as a tool to first and foremost simulate runoff, issues can arise when simulating other fluxes and state variables, such as soil moisture or evapotranspiration. New approaches have been developed to facilitate the incorporation of remotely sensed data to support watershed scale studies [14]. Of particular promise are data assimilation (DA) techniques adopted from the atmospheric science community, which have been increasingly applied to watershed hydrology studies [15–17]. However, the improvements that can be potentially conferred by DA have limitations. DA has difficulty

in improving streamflow performance under high flow conditions [15,16] because runoff production is largely decoupled from the control of soil moisture under these circumstances. In addition, SWAT has some structural issues related to how soil moisture is accounted for that limits the benefits of DA of root zone soil moisture (RZSM) in this model. For example, the authors in reference [17] used DA to incorporate RZSM into SWAT and achieved worse results than open loop simulations. This is because the physics of the SWAT model without modification are not sufficiently complicated to account for vertical coupling between different soil layers. Despite these issues, soil moisture remains an important control on surface runoff production. One of the most important parameters within SWAT is the Curve Number (CN2), which is initialized based on the moisture content within soils. Therefore, finding a way of leveraging soil moisture to support more realistic modeling of streamflow remains important.

Another approach that provides a more holistic prospective is a mass balance accounting of the overall water budget. This method has yielded meaningful insights particularly at the regional and watershed scales (e.g., [18–20]). In reference [21], it is indicated that inter-seasonal and inter-annual variations in surface water storage volumes, as well as their impact on precipitation (P), evapotranspiration (ET), surface water storage (S), and runoff (Q), are not well understood. There remains a fundamental lack of knowledge, both in terms of spatial and temporal scales, regarding the hydrologic processes that influence each of the terms of the basic hydrologic equation. Incorporation of multiple observations (both in situ and remotely sensing) into model calibration can force modeling to be based on more realistic parameter selection. Therefore, the objective of this study is to demonstrate whether diverse remote sensing observations can improve simulated SWAT streamflow in eight Great Plains watersheds.

## 2. Watersheds Examined

Eight, moderate-sized (832 to 4892 square km) watersheds were examined (Table 1; Figure 1). Basins generally have a dendritic drainage pattern with a rounded shape, except for Chickaskia (CH) and Ninnescah (NI), which are elongated. Bird Creek (BC), CH, Little Arkansas (LA), and Little Nemaha (LN) flow in general toward the southeast. Black Vermillion (BV) drainage is oriented southwest and Walnut (WN) toward the south. Mill Creek (MC) and NI flow toward the east. The SWAT model is subdivided into subbasins as computational units. To enhance inter-comparability of the results, the number of subbasins was set as consistently as possible. The eight basins had subdued topography typical of the Great Plains region. Overall relief varied between 130 to 313 m in the examined watersheds (Table 1). In terms of soils, most watersheds were dominated by some variants of loam within the top layer that roughly correspond with the upper root zone. The only exception was NI, where loamy sand was the most abundant texture. Land use/land cover in five watersheds was dominated by agricultural activity (BV, CH, LA, LN, NI). BC, MC, and WN also had significant rangeland and grasses.

**Table 1.** Watershed characteristics.

| Basin | Size (sq. km.) | Subbasins | Elevation (m) | Dominant Soil Texture | Dominant Land Cover |
|---|---|---|---|---|---|
| Bird Creek (BC) | 2360 | 31 | 177 to 403 | Loam | Rangeland/Grass |
| Black Vermillion (BV) | 1071 | 31 | 338 to 468 | Clay Loam | Agricultural |
| Chickaskia (CH) | 4892 | 33 | 295 to 608 | Silt Loam | Agricultural |
| Little Arkansas (LA) | 3402 | 33 | 409 to 544 | Silt Loam | Agricultural |
| Little Nemaha (LN) | 2061 | 31 | 274 to 444 | Clay | Agricultural |
| Mill Creek (MC) | 832 | 29 | 291 to 488 | Silt Clay Loam | Rangeland/Grass |
| Ninnescah (NI) | 2049 | 35 | 446 to 637 | Loamy Sand | Agricultural |
| Walnut (WN) | 4855 | 33 | 330 to 512 | Silt Loam | Rangeland/Grass |

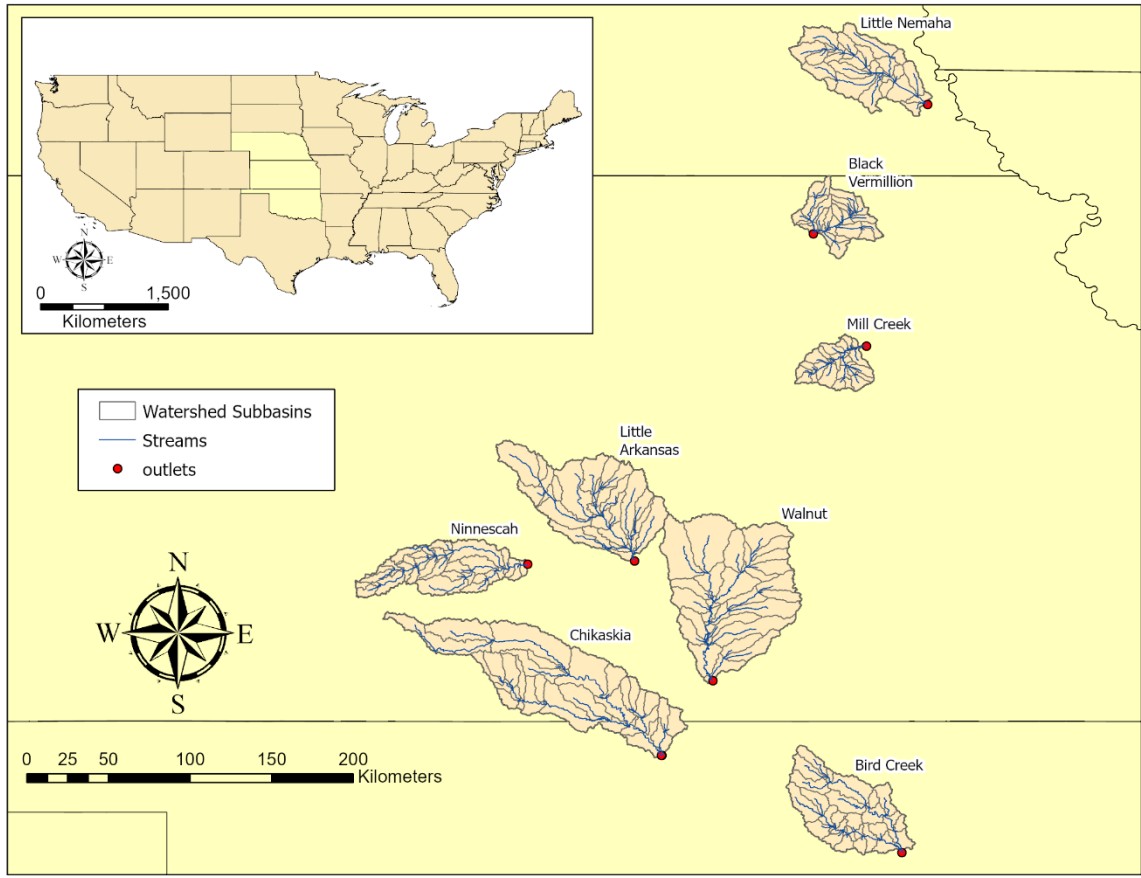

**Figure 1.** Locality map illustrating the position of the eight examined watersheds.

## 3. Datasets Used

### 3.1. SWAT Model Input

The SWAT model incorporates landscape information about elevation, soils, and land use/land cover. Elevation data were derived from the National Map Download service from the United States Geological Survey. Tiles of this seamless product were downloaded in an ArcGrid format, with a spatial resolution of 1 arc-second. For soils, the Digital General Soil Map of the United States or STATSGO2 (United States (US) Department of Agriculture, Washington, DC, USA) developed by the National Cooperative Soil Survey, was selected. This product was downloaded from the Natural Resources Conservation Service Geospatial Data Gateway in polygon shapefile format and it has an inherent 1:250,000 spatial scale. Finally, land use/land cover data came from the 2011 National Land Cover Data Set, accessed through the Natural Conservation Service Geospatial Data Gateway. This product was developed by the Multi-Resolution Land Characteristics Consortium. It was obtained in a GeoTIFF format with a spatial resolution of 30 m.

Hydrometeorological data (daily precipitation and temperature) were obtained from the PRISM Climate Group at Oregon State University. This product has a 4 km spatial resolution covering all of the continental United States (CONUS). To support execution of the SWAT model, daily PRISM data for all SWAT subbasins was averaged using zonal statistics based on the intersection of PRISM grid cell with each subbasin. In addition, annual basin-wide averages for precipitation were obtained for all eight basins by simple averaging.

### 3.2. Other Soft Data

For RZSM, the SMERGE 2.0 product (US National Aeronautics and Space Administration, NASA, Washington, DC, USA) was selected [22]. This product provided particularly robust results in the Great

Plains region and reflects a product that blends equally remote sensing and land surface model datasets. SMERGE 2.0 is available at a daily time step and has a 0.125-degree spatial resolution. Like with PRISM precipitation data, SMERGE was averaged for each of the eight basins on an annual basis (1992 to 2018). Unlike PRISM data anomalies, not raw volumetric data were used for RZSM.

Three ET products were applied to this study (Moderate Resolution Imagining Spectrometer, MODIS16A2v5, US NASA Earth Observing System Data and Information System (EOSDIS) Land Processes Distributed Active Archive Center (DAAC), [23]; Simplified Surface Energy Balance, SSEBopv4, US Geologic Survey, Center for Integrated Data Analytics, Middleton, Wisconsin [24]; Global Land Evaporation: the Amsterdam Model, GLEAMv3.3a, Vrije Universiteit Amsterdam, The Netherlands [25]). The MODIS product was obtained in a monthly HDF file with a 1 km resolution. This dataset was extracted into a raster layer and zonal statistics tools were utilized to obtain the average of the pixels intersecting with the watershed outline. The same method was applied to SSEBop, which was obtained in a GeoTiff raster format with a 0.009-degree spatial resolution in monthly files. GLEAM was available in netCDF files in a grid of 0.25 × 0.25 degrees with a monthly temporal resolution. The values of the grids with centroids within the watershed were extracted and summed to obtain the average. Values were summed to calculate an annual estimate of ET (2016–2018) for the eight examined watersheds.

Total terrestrial water was estimated from the NASA Gravity Recovery and Climate Experiment (GRACE) using the GRCTellus JPL-Mascons dataset [26,27]. This product combined monthly gravity solutions from GRACE and GRACE-FO, as determined from the JPL RL06Mv2 mascon solution with the coastline resolution improvement filter. The GRACE product was available in a monthly netCDf file (missing values exist) at a 0.5-degree spatial resolution. The average of intersecting GRACE grids with the watershed was summed to obtain an annual estimate (2015–2018) of total terrestrial water change.

Finally, to constrain an interception-related SWAT parameter, the MOD15A2H Terra version 006 combined Leaf Area Index (LAI) and Fraction of Photosynthetically Active Radiation (US NASA EOSDIS Land Processes DAAC, Sioux Falls, ND, USA) was used [28]. This product had a 500 m spatial resolution and was an eight-day composite dataset based on the best available from acquisitions from each period. Values were aggregated to obtain a basin-wide estimate of LAI.

## 4. Methodology

### 4.1. SWAT Model Setup

In SWAT, the automatic watershed delineation tool was used to define the stream network and number of subbasins within a watershed. Subbasin number was based on the area of the watershed present upstream of the beginning point for each tributary channel. Within each subbasin, water balance calculations were based on the aerially weighted proportions of unique combinations of soil and land use, referred to as hydrologic response units (HRU). Each HRU had a unique Curve Number (CN), adjusted for antecedent moisture conditions, which was used to determine infiltration and surface runoff within each subbasin. Another component of the SWAT model that enhanced its ability to calculate the water balance within each subbasin was the calculation of daily potential evapotranspiration values using the Priestley–Taylor method [29]. SWAT does not consider the spatial location of HRUs within each subbasin, and consequently, this is why SWAT is not considered a fully distributed model. Excessive runoff generated within each subbasin was conceptually routed as overland flow. Once overland flow water intersected a stream reach or channel, water was routed downstream using the variable storage method [30].

To facilitate autocalibration of the SWAT model, the stand-alone SUFI-2 autocalibration [31] routine was utilized. Of the autocalibration programs available for the SWAT model, SUFI-2 converges on an optimal solution with a relatively small number of executed simulations (500 to 1000 model runs; [32,33]) and was ran at a daily time step. In addition, SUFI-2 provided an estimate of parameter sensitivity. Note that HRU parameter values were averaged at the subbasin level. Only highly sensitive parameters (HSP; *p*-value < 0.02) were varied after the first two global simulations executed, which are described below.

To evaluate model performance, standard objective measures were used, including the mass balance error (MBE) and Nash–Sutcliffe efficiency coefficients (NS). To collapse these metrics into one measure, all model results were evaluated based on the Relative Performance Scale [34]. This combined metric was based on the criterion of reference [35] (see Table 2). To calculate the RPS, both the MBE and NS were translated into a single RPS metric. For example, if a simulation has a NS = 0.75 and MBE of 15%, these values constituted provisional RPS values of 3.00 and 2.00, respectively. To be conservative, the lower provisional RPS value was always selected so that in this example, the final RPS value assigned was 2.00. The best model run for each simulation type was evaluated with a single RPS score to facilitate inter-comparison of results.

**Table 2.** The Relative Performance Scale (RPS).

| Description | Nash Sutcliffe (NS) | Mass Balance Error | Relative Performance Scale (RPS) |
|---|---|---|---|
| Perfect | 1.00 | 0% | 4.00 |
| Very Good | 0.75 | 10% | 3.00 |
| Good | 0.65 | 15% | 2.00 |
| Satisfactory | 0.50 | 25% | 1.00 |
| Unacceptable | <0.50 | >25% | <1.00 |

*4.2. Simulation Series*

Three series of model runs were executed in this study and include: (1) global simulations (1995 to 2015); (2) individual year-by-year models runs for each year between (1995 to 2015); (3) final calibration year-by-year simulations (2016 to 2018). For all series, a three- to four-year warm up period was executed to initialize SWAT.

4.2.1. Global Simulation Series

For global simulations, one RPS value was calculated for the entire simulation period (1995 to 2015) in each watershed; shorter for MC (2005 to 2015). This simulation series consisted of two model runs. The initial simulation was referred to as Base_Q. In this model run, there were no constraints on parameters values, except for the outer bounds established by reference [29]. Parameter value ranges for the Base_Q are presented in Table 3.

The next type of global simulation consisted of iterative model runs, which constrained parameters to improve objective metrics and was referred to as IT_Q. Model parameters were limited in two ways: (a) using *a priori* data to set CANMX and ALPHA_BF and (b) examining Dotty plots (Figure 2) to identify limits for optimal performance for variable HSP. The CANMX parameter was set by using MODIS_LAI product and the following equation [36].

$$S_{Max} = f \log (1 + LAI) \tag{1}$$

where $S_{Max}$ was the maximum water storage within the canopy, $f$ was a specific factor dependent upon vegetation type, and LAI was determined from MODIS_LAI product (MOD15A2). ALPHA_BF was determined using the baseflow program from reference [37] and was set within a factor of two of the calculated value. Parameter sensitivity was examined and HSP were identified (Table 4). These parameters can be divided into two groups (variable and non-variable). Examination of Dotty Plots

can further constrain variable, HSP values. While some of the parameters lack an optimal range of values (Figure 2a), others do not (Figure 2b). An iterative approach was used in adjusting parameters with optimal values to yield better performance. HSP specifically, in all basins, the tightening of the range of CH_K2 improved results. Additional iterations in BC and WN focused on CH_N2; in NI, with CN2; in WN, on OV_N. The tightening of variable HSP values had a beneficial impact on the final IT_Q model executed. From this simulation, the values of non-sensitive ($p > 0.02$) and non-variable HSP are set from the optimum parameter values calculated (Table 5). Only variable, HSP (Table 6) were left unconstrained in subsequent modeling series.

### 4.2.2. Individual Year-By-Year Series

Individual year-by-year model runs between 1995 to 2015 were executed. In this modeling series, objective results were obtained for each year (n = 21). All variable, HSP were correlated with annual SMERGE 2.0 RZSM anomalies and raw PRISM precipitation. Note that years with unacceptable RPS values were omitted from this analysis. Correlation values based on this modeling series weare presented in Table 7. The range and average for variable, HSP are shown in Table 8. Only parameters with a correlation (r) that exceeds 0.5 were considered in the third modeling series described next.

**Table 3.** Base_Q parameter ranges for all basins.

| Parameter | Name | Low | High |
|---|---|---|---|
| CN2 | Initial SCS runoff curve number for moisture condition II | 35 | 95 |
| ALPHA_BF | Baseflow Alpha Factor | 0 | 1 |
| GW_DELAY | Groundwater delay time (days) | 30 | 450 |
| CH_N2 | Manning's "n" value for the main channel | 0 | 0.3 |
| CH_K2 | Effective hydraulic conductivity in main channel alluvium (mm/h) | 0 | 500 |
| CH_N1 | Manning's "n" value for the tributary channels | 0 | 0.3 |
| CH_K1 | Effective hydraulic conductivity in tributary channel alluvium (mm/h) | 0 | 300 |
| OV_N | Manning's "n" value for overland flow | 0.01 | 0.6 |
| SURLAG | Surface runoff lag coefficient | 1 | 34 |
| GWQMN | Threshold depth of water in the shallow aquifer required for return flow to occur (mm $H_2O$) | 0 | 5000 |
| SOL_AWC | Available water capacity of the soil layer (mm $H_2O$/mm soil) | −0.2 | 0.4 |
| ESCO | Soil evaporation compensation factor | 0 | 1 |
| GW_REVAP | Groundwater "revap" coefficient | 0.02 | 0.2 |
| REVAPMN | Threshold depth of water in the shallow aquifer for "revap" or percolation to the deep aquifer to occur (mm $H_2O$) | 0 | 500 |
| CANMX | Maximum canopy storage (mm $H_2O$) | 0 | 100 |
| EPCO | Plant uptake compensation factor | 0 | 1 |
| SFTMP | Snowfall temperature (°C) | −5 | 5 |
| SMTMP | Snow melt base temperature (°C) | −5 | 5 |
| SMFMX | Melt factor for snow on June 21 (mm $H_2O$/°C-day) | 0 | 10 |
| SMFMN | Melt factor for snow on Dec 21 (mm $H_2O$/°C-day) | 0 | 10 |
| TIMP | Snow pack temperature lag factor | 0.01 | 1 |
| SOL_K | Saturated hydraulic conductivity (mm/h) | −0.8 | 0.8 |
| SOL_BD | Moist bulk density (g/cm$^3$) | −0.5 | 0.6 |

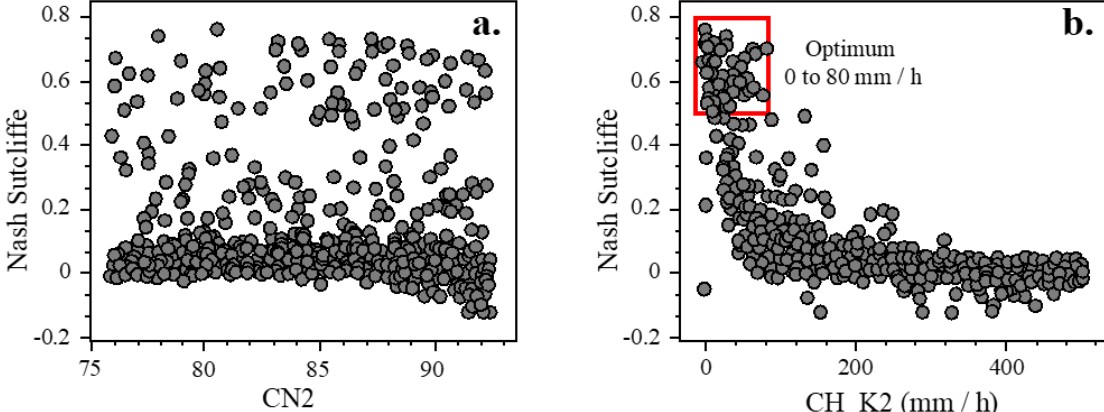

**Figure 2.** Dotty plots for BV basin. (**a**) NS versus CN2 and (**b**) NS versus CH_K2 parameter value with zone of optimum performance indicated.

**Table 4.** Highly sensitive parameters (HSP) from IT_Q simulation.

| Basin | Variable Parameters | Non-Variable Parameters |
|---|---|---|
| BC | CN2, ALPHA_BF, CH_N2, OV_N | SOL_BD, GWQMN, ESCO |
| BV | CN2, CH_K2, OV_N, ESCO | CH_N2, ALPHA_BF, SOL_AWC |
| CH | CN2, CH_N2, CH_K2, OV_N, ESCO | ALPHA_BF |
| LA | CN2, CH_N2, CH_K2, OV_N, SOL_BD | ALPHA_BF, SMTMP |
| LN | CN2, OV_N, SOL_AWC, ESCO, SOL_BD | SMFMN |
| MC | CN2, CH_K2, ESCO | CH_N2, OV_N, SMTMP |
| NI | CN2, CH_N2, CH_K2, OV_N, SOL_BD | SMTMP |
| WN | CN2, CH_K2, OV_N, ESCO | ALPHA_BF, CH_N2, SOL_AWC |

**Table 5.** Highly sensitive parameters (HSP) from IT_Q simulation.

| Parameter | BC | BV | CH | LA | LN | MC | NI | WN |
|---|---|---|---|---|---|---|---|---|
| ALPHA_BF | | 0.112 | 0.0444 | 0.073 | 0.0427 | 0.012 | 0.033 | 0.0572 |
| GW_DELAY | 252 | 136 | 43.6 | 81.4 | 105 | 443 | 166 | 189 |
| CH_N2 | | 0.255 | | | 0.221 | 0.267 | | 0.065 |
| CH_K2 | 39.0 | | | | 3.20 | | | |
| CH_N1 | 0.184 | 0.0091 | 0.300 | 0.084 | 0.049 | 0.254 | 0.044 | 0.242 |
| CH_K1 | 11.0 | 297 | 239 | 83.8 | 73.9 | 275 | 123 | 190 |
| OV_N | | | | | | 0.582 | | |
| SURLAG | 6.16 | 12.6 | 16.3 | 9.03 | 4.84 | 7.02 | 20.5 | 3.92 |
| GWQMN | 1967 | 2342 | 3517 | 2287 | 117 | 2167 | 4707 | 4552 |
| SOL_AWC | 0.268 | 0.370 | 0.352 | 0.0031 | | 0.384 | −0.125 | 0.114 |
| ESCO | 0.043 | | | 0.865 | | | 0.186 | |
| GW_REVAP | 0.118 | 0.056 | 0.077 | 0.068 | 0.159 | 0.191 | 0.177 | 0.026 |
| REVAPMN | 353 | 66.2 | 337 | 468 | 492 | 19.2 | 138 | 72.7 |
| CANMX | 0.514 | 0.405 | 0.396 | 0.255 | 0.322 | 0.368 | 0.470 | 0.691 |
| EPCO | 0.863 | 0.435 | 0.736 | 0.743 | 0.058 | 0.894 | 0.254 | 0.346 |
| SFTMP | −2.38 | −4.25 | −4.32 | 2.33 | −0.245 | −0.735 | −1.56 | 0.905 |
| SMTMP | −0.845 | 1.24 | 2.96 | 0.975 | 0.765 | 4.51 | 4.98 | −2.69 |
| SMFMX | 0.095 | 6.46 | 0.965 | 7.46 | 1.40 | 3.38 | 9.49 | 9.70 |
| SMFMN | 5.45 | 0.665 | 6.78 | 0.305 | 5.65 | 4.95 | 4.29 | 7.57 |
| TIMP | 0.869 | 0.579 | 0.587 | 0.795 | 0.735 | 0.156 | 0.671 | 0.689 |
| SOL_K | 0.294 | −0.418 | 0.257 | 0.036 | −0.401 | −0.310 | −0.015 | 0.434 |
| SOL_BD | −0.142 | 0.339 | −0.092 | | | −0.096 | | 0.281 |

**Table 6.** IT_Q SWAT ranges for parameters that are variable and HSP.

| Parameter | BC | BV | CH | LA | LN | MC | NI | WN |
|---|---|---|---|---|---|---|---|---|
| CN2 | 60–84 | 76–92.4 | 72–88 | 68–84 | 68–84 | 70–88 | 45.9–80 | 76–90.3 |
| ALPHA_BF | 0.035–0.139 | | | | | | 0–0.3 | |
| CH_N2 | 0.015–0.04 | | 0–0.3 | 0–0.3 | | 0–40 | 0–40 | 10–30 |
| CH_K2 | | 0–20 | 0–40 | 0–40 | 0.01-0.6 | | 0.01–0.6 | 0.4–0.6 |
| OV_N | 0.01–0.60 | 0.01–0.6 | 0.01–0.6 | 0.01–0.6 | −0.2–0.4 | | | |
| ESCO | | 0–1.0 | 0–1.0 | | 0–1.0 | 0–1.0 | | 0–1.0 |
| SOL_BD | | | | −0.5–0.6 | −0.5–0.6 | | −0.5–0.6 | |

**Table 7.** Parameter correlation (r) versus PRISM precipitation and SMERGE 2.0 root zone soil moisture anomalies based on individual year (1992 to 2015) runs.

| SMERGE 2.0 | | | | | | | | |
|---|---|---|---|---|---|---|---|---|
| Parameter | BC | BV | CH | LA | LN | MC | NI | WN |
| CN2 | **0.618** | 0.457 | **0.658** | 0.240 | 0.361 | **0.727** | **0.791** | **0.725** |
| ALPHA_BF | 0.266 | | | | | | | |
| CH_N2 | 0.114 | | −0.116 | 0.290 | | | 0.399 | 0.179 |
| CH_K2 | | 0.191 | 0.324 | 0.345 | | 0.129 | −0.150 | 0.437 |
| OV_N | 0.342 | 0.246 | 0.015 | −0.437 | −0.237 | | 0.231 | |
| SOL_AWC | | | | | 0.048 | | | |
| ESCO | | −0.301 | | | −0.251 | −0.707 | | |
| SOL_BD | | | | −0.450 | −0.075 | | 0.041 | |
| PRISM | | | | | | | | |
| Parameter | BC | BV | CH | LA | LN | MC | NI | WN |
| CN2 | 0.462 | **0.515** | 0.499 | 0.293 | 0.347 | **0.662** | **0.539** | 0.440 |
| ALPHA_BF | 0.297 | | | | | | | |
| CH_N2 | 0.031 | | −0.121 | −0.052 | | | 0.329 | 0.321 |
| CH_K2 | | **0.512** | 0.346 | **0.577** | | −0.124 | −0.255 | 0.227 |
| OV_N | 0.201 | 0.172 | −0.092 | −0.493 | −0.433 | | 0.033 | |
| SOL_AWC | | | | | −0.247 | | | |
| ESCO | | −0.384 | | | **−0.572** | **−0.651** | | |
| SOL_BD | | | | −0.336 | −0.219 | | 0.058 | |

HCP with r > 0.5 are in bold.

**Table 8.** Range and average (in parentheses) of HSP from individual year (1992 to 2015) runs.

| Parameter | BC | BV | CH | LA |
|---|---|---|---|---|
| CN2 | 67.1 to 83.9 (76.7) | 77.9 to 89.7 (84.5) | 75.4 to 85.6 (81.5) | 69.0 to 83.2 (75.8) |
| ALPHA_BF | 0.035 to 0.127 (0.092) | | | |
| CH_N2 | 0.016 to 0.060 (0.033) | | 0.044 to 0.206 (0.104) | 0.065 to 0.180 (0.112) |
| CH_K2 | | 1.5 to 10.5 (7.7) | 4.7 to 15.5 (10.1) | 2.6 to 17.7 (9.6) |
| OV_N | 0.022 to 0.591 (0.365) | 0.101 to 0.441 (0.303) | 0.206 to 0.572 (0.381) | 0.274 to 0.585 (0.396) |
| ESCO | | 0.456 to 0.868 (0.612) | | |
| SOL_BD | | | | −0.431 to 0.443 (0.008) |

| Parameter | LN | MC | NI | WN |
|---|---|---|---|---|
| CN2 | 69.8 to 82.8 (77.8) | 70.7 to 85.2 (77.7) | 47.9 to 76.3 (68.0) | 76.6 to 90.1 (84.7) |
| CH_N2 | | | 0.018 to 0.283 (0.164) | |
| CH_K2 | | 4.0 to 23.1 (11.5) | 2.2 to 24.0 (13.4) | 10.4 to 25.1 (16.8) |
| OV_N | 0.301 to 0.581 (0.410) | | 0.211 to 0.578 (0.387) | 0.407 to 0.591 (0.519) |
| SOL_AWC | −0.172 to 0.362 (0.089) | | | |
| ESCO | 0.583 to 0.959 (0.756) | 0.171 to 0.981 (0.639) | | |
| SOL_BD | −0.422 to 0.588 (0.0175) | | −0.389 to 0.523 (0.065) | |

### 4.2.3. Final Calibration Year-By-Year Series

Information from the two prior modeling series was leveraged to improve calibration in the final year-by-year calibration series (2016 to 2018). Unacceptable objective metrics (RPS < 1.00 for Sens_Q) were used to omit the following basin–year combinations (BC-2016, BC-2018, NI-2017, and WN-2017).

Four model runs that were executed in this series, which include: (a) Global_Q that applied IT_Q parameter values (Tables 4 and 6) on a year-by-year basis (i.e., 2016 Global_Q); (b) Sens_Q in which variable, HSP (Table 8) were set between the range observed during the 1995 to 2015 runs; (c) SMERGE_Parameter (i.e., 2016 SMERGE_CN2); (d) PRISM_Parameter (i.e., 2016 PRISM_CN2). For the parameter-based model runs variable, HSP were set at ±10% of the average value obtained between 1995 to 2015 (Table 8), except for the highly correlated parameters (HCP). HCP values were calculated using the SMERGE 2.0 RZSM anomaly or raw PRISM precipitation for the examined year (i.e., 2016; Table 9) using the 1995 to 2015 regression relationship. The parameter range for HCP was set at ±10% of the calculated value.

### 4.3. Mass Balance Calculations

Streamflow (Q) simulated from year-by-year series (2016 to 2018) was compared against USGS gauge observed streamflow (with a nominal ±10% error). The range of simulated Q were produced by extracting all simulations in a model run that yielded acceptable results (RPS ≥ 1.00). Streamflow was calculated based on mass balance within each basin based on:

$$Q_{Calculated} = P - \Delta S - ET \tag{2}$$

where P was the annual average PRISM precipitation value within a watershed; ΔS was the change in annual terrestrial water determined from the GRACE product; ET was evapotranspiration and was estimated with three products (MODIS16A2v5; GLEAM v.3.3a; SSEBop v.4). A nominal 10% error was applied to calculated Q values. Note that LN-2018 was omitted in the mass balance analysis because of incomplete observed USGS streamflow data at the end of 2018.

## 5. Results

### 5.1. SWAT Simulations

The initial global series included Base_Q and IT_Q simulations. Only BV and CH have acceptable Base_Q simulations. The IT_Q results are dramatically better. Only MC was not satisfactory. BC, LA, NI, and WN were satisfactory to good, CH good to very good, and BV and NH exceeded the threshold for a very good simulation. Figure 3 combines the results from all eight basins into box plots. The average for the Base_Q model runs had an RPS value of 0.781, which was unacceptable. The iterative approach IT_Q improved objective results, with an average RPS of 1.886. T-test comparison between Base_Q and IT_Q simulations yielded a significant difference (based on t-test results) between the means of these model runs (p value = 0.0058). This comparison shows how constraining parameters improved model performance.

The final calibration year-by-year series had four types of model runs that included Global_Q, Sens_Q, SMERGE_Parameter, and PRISM_Parameter (Figure 4). Global_Q, which was based on IT_Q parameter values, had an average RPS of 1.282, considered satisfactory to good. In BC and NI, performance was unsatisfactory for all years (2016 to 2018). In other basins, simulations varied greatly on a year-by-year basis. In BV, results ranged from unsatisfactory to very good and in LA and MC, from unsatisfactory to good. LN recorded results between satisfactory and very good. In CH and WN, performance ranges between unsatisfactory to satisfactory.

The three other model series (Sens_Q, SMERGE_Parameter, and PRISM_Parameter) leveraged the results from the individual year-by-year simulation series (1995 to 2015) to constrain parameter values. These series yielded average RPS values (2.032 to 2.623), which ranged from good to very good in

terms of performance. Notable improvements in the three-model series over Global_Q were noted for the following basin–year combination, which recorded an over 2.00 increase in RPS values (BV-2018, CH-2018, LA-2018, MC-2016, and NI-2016). Table 10 provided a summary of t-test results for these simulation series. The Global_Q model run had a statistically significant difference compared with the three other simulation series. Conversely, Sens_Q, SMERGE_Parameter, and PRISM_Parameter did not differ significantly between each other (Table 10). These results demonstrated a range of optimal solutions achieved with differing parameter values—a prime example of how equifinality can limit the utility of hydrologic simulations.

**Table 9.** Highly correlated parameter values for final calibration year-by-year (2016 to 2018) runs.

| Basin | Year | Product | CN2 | CH_K2 | ESCO |
|-------|------|---------|-----|-------|------|
| BC | 2017 | SMERGE 2.0 | 78.5 | | |
| BC | 2017 | PRISM | 82.5 | | |
| BV | 2016 | SMERGE 2.0 | 84.4 | 6.7 | |
| BV | 2016 | PRISM | 84.2 | 6.6 | |
| BV | 2017 | SMERGE 2.0 | 81.0 | 4.1 | |
| BV | 2017 | PRISM | 80.2 | 3.5 | |
| BV | 2018 | SMERGE 2.0 | 81.8 | 4.7 | |
| BV | 2018 | PRISM | 83.3 | 5.8 | |
| CH | 2016 | SMERGE 2.0 | 81.2 | | |
| CH | 2016 | PRISM | 81.6 | | |
| CH | 2017 | SMERGE 2.0 | 80.5 | | |
| CH | 2017 | PRISM | 80.1 | | |
| CH | 2018 | SMERGE 2.0 | 80.1 | | |
| CH | 2018 | PRISM | 83.7 | | |
| LA | 2016 | SMERGE 2.0 | | 12.8 | |
| LA | 2016 | PRISM | | 17.2 | |
| LA | 2017 | SMERGE 2.0 | | 7.1 | |
| LA | 2017 | PRISM | | 7.6 | |
| LA | 2018 | SMERGE 2.0 | | 8.0 | |
| LA | 2018 | PRISM | | 15.7 | |
| LN | 2016 | SMERGE 2.0 | | | 0.824 |
| LN | 2016 | PRISM | | | 0.798 |
| LN | 2017 | SMERGE 2.0 | | | 0.726 |
| LN | 2017 | PRISM | | | 0.761 |
| LN | 2018 | SMERGE 2.0 | | | 0.798 |
| LN | 2018 | PRISM | | | 0.810 |
| MC | 2016 | SMERGE 2.0 | 79.5 | | 0.693 |
| MC | 2016 | PRISM | 85.1 | | 0.943 |
| MC | 2017 | SMERGE 2.0 | 77.5 | | 0.602 |
| MC | 2017 | PRISM | 84.3 | | 0.907 |
| MC | 2018 | SMERGE 2.0 | 74.1 | | 0.453 |
| MC | 2018 | PRISM | 81.1 | | 0.764 |
| NI | 2016 | SMERGE 2.0 | 62.4 | | |
| NI | 2016 | PRISM | 67.1 | | |
| NI | 2018 | SMERGE 2.0 | 65.0 | | |
| NI | 2018 | PRISM | 78.9 | | |
| WN | 2016 | SMERGE 2.0 | 85.0 | | |
| WN | 2016 | PRISM | 89.2 | | |
| WN | 2017 | SMERGE 2.0 | 81.3 | | |
| WN | 2017 | PRISM | 82.1 | | |
| WN | 2018 | SMERGE 2.0 | 80.3 | | |
| WN | 2018 | PRISM | 85.5 | | |

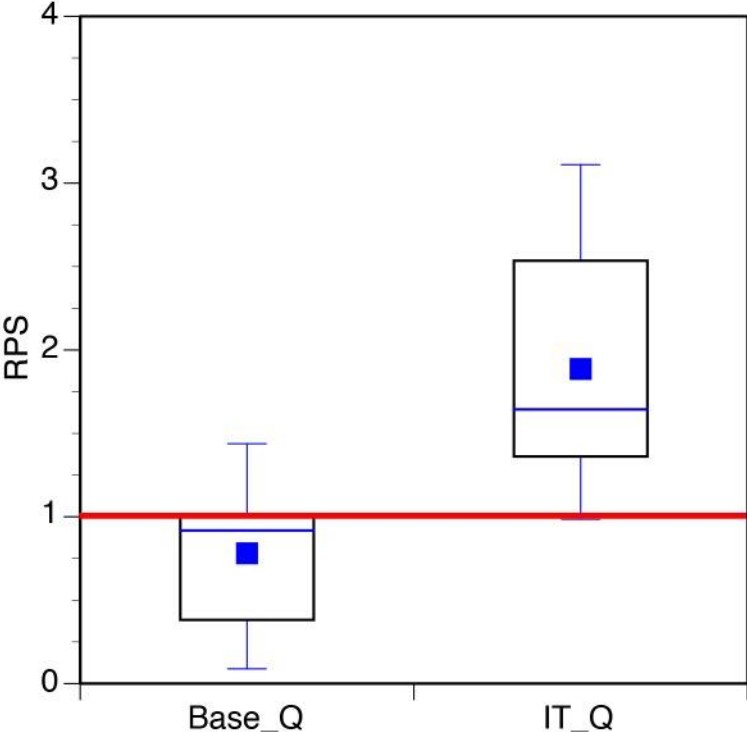

**Figure 3.** Combined RPS value from global simulation series (1995 to 2015). Red line represents an RPS = 1.00, which is the threshold for an acceptable simulation.

**Table 10.** T-test comparison of mean RPS values from final year-by-year (2016 to 2018) runs.

| Model Run #1 | Model Run #2 | *p* Value | Degree of Significance |
|---|---|---|---|
| Global_Q | Sens_Q | 0.0001 | Highly Significant |
| Global_Q | SMERGE_Parameter | 0.0001 | Highly Significant |
| Global_Q | PRISM_Parameter | 0.0309 | Significant |
| Sens_Q | SMERGE_Parameter | 0.5269 | Not Significant |
| Sens_Q | PRISM_Parameter | 0.0659 | Not Significant |
| SMERGE_Parameter | PRISM_Parameter | 0.1452 | Not Significant |

*5.2. Mass Balance Comparisons*

To avoid the equifinality constraint, calculated Q, which used P, ET, and ΔS, was examined. Observed, SWAT simulated, and calculated Q are compared in Figures 5–10 for the years 2016 to 2018. Similar results were obtained in most the watersheds, as discussed below. Note that BC-2016, BC-2018, and NI-2017 were omitted because no acceptable simulations were obtained during these years. LN-2018 was omitted because complete observational data were not available for 2018.

Calculated Q for all three products compared poorly against USGS gauge observed Q and in general, grossly overestimated this value. MODIS16A2v5 nearly always generated a highly inflated calculated Q value. GLEAM v.3.3a and SSEBop v.4 exhibited more variability but still had the general tendency to produce an overestimate of calculated Q. GLEAM v.3.3a (BV-2017, LA-2017, MC-2016, and MC-2017; Figures 5c, 7b and 9a,b) and SSEBop v.4 (CH-2016, LA-2017, and NI-2017; Figure 6a, Figure 7b, and Figure 10a) underestimated observed Q four and three times, respectively. Out of the 20 acceptable basin and year combinations examined, only two of the calculated Q based on MODIS16A2v5 (MC-2016 and MC-2017; Figure 9a,b) and four based on GLEAM v.3.3a (BV-2017, CH-2017, MC-2016, and MC-2017; Figures 5a, 6b and 9a,b) overlapped with the nominal observed Q band. SSEBop v.4 calculated Q matched slightly better with the nominal observational band, with eight basin and year combinations (BC-2017, CH-2017, LA-2016, LN-2016, MC-2016, MC-2017, WN-2016, and WN-2017; Figure 5a, Figure 6b, Figure 7a, Figure 8a, Figure 9a,b and Figure 10c).

Comparing ET products, generally, MODIS16A2v5 yielded the highest calculated Q and SSEBop v.4 the lowest calculated Q, with GLEAM v.3.3a in the middle. Exceptions to this rule included BV-2016, BV-2017, LA-2017, MC-2016, MC-2017, and WN-2017 (Figure 5b,c, Figures 7b and 9a,b), which had GLEAM v.3.3a as the lowest calculated Q value. For MC-2016 (Figure 9a), SSEBop v.4 had the highest value.

In terms of the final calibration year-by-year series, the simulated Q much better matched with the USGS gauge observed Q. Sens_Q generally had a wider range of simulated Q values compared with either SMERGE_Parameter or PRISM_Parameter, in which parameter values were more constrained. The exception to this is in BV and MC, which had two HPCs (Table 9) that yielded a range of simulated Q similar to that of Sens_Q. All but one (WN-2017) Sens_Q runs had acceptable RPS values and overlapped with the nominal ±10% error band associated with USGS gauge observed Q. Sixteen of the SMERGE_Parameter and only twelve PRISM_Parameter model runs overlapped with the nominal observational band. SMERGE_Parameter models had two overestimates (CH-2016, CH-2018; Figure 6a,c) and two underestimates (LN-2017; WN-2017; Figure 8b) relative to observed Q. PRISM_Parameter had six overestimates (BC-2017, CH-2016, CH-2018, NI-2018, WN-2016, and WN-2018; Figure 5a, Figure 6a,c and Figure 10b–d) and two underestimates (LA-2017 and LN-2017; Figures 7b and 8b). In addition, one SMERGE_Parameter and four PRISM_Parameter model runs had unacceptable RPS values, indicating that parameter constraints on occasion degraded model performance.

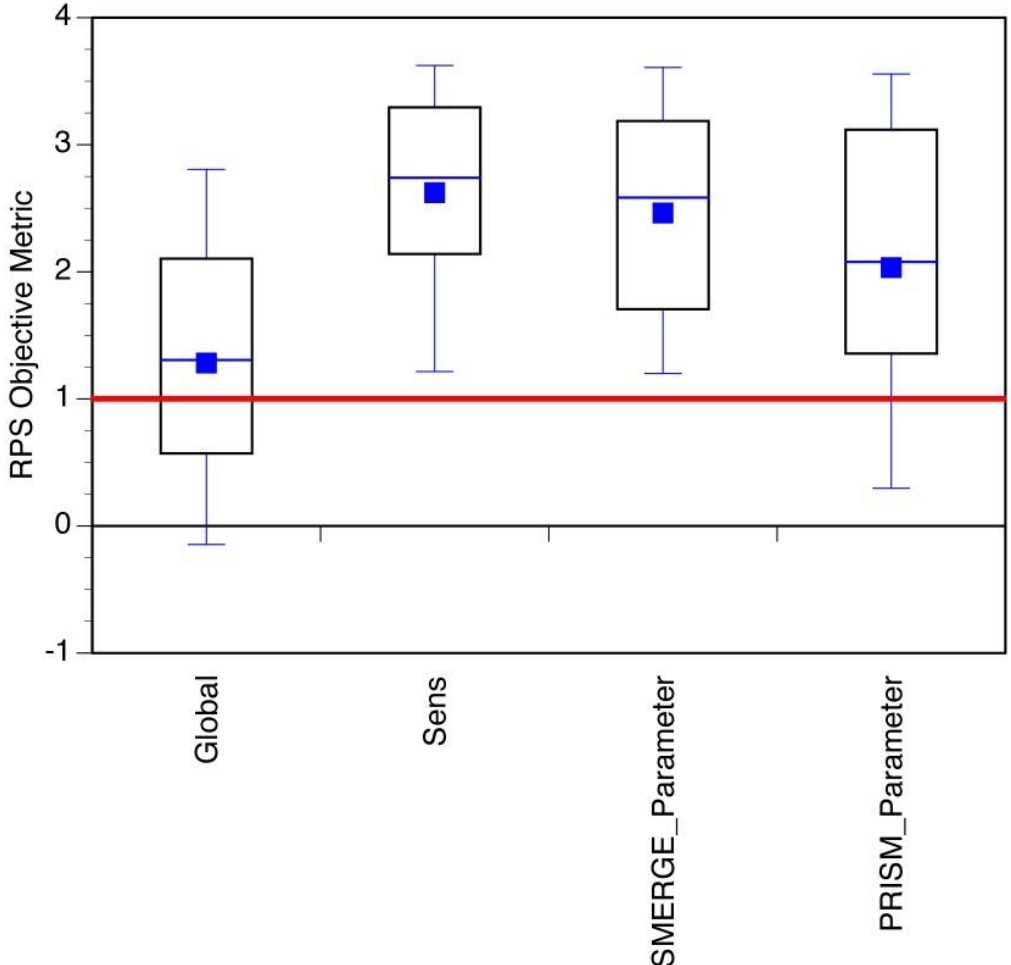

**Figure 4.** Combined RPS value from final year-by-year series (2016 to 2018). Red line represents an RPS = 1.00, which is the threshold for an acceptable simulation.

## 6. Discussion

This work is a fundamental example of how constraining parameter values in a hydrologic model can improve objective performance measures during autocalibration. Even so the issue of equifinality remains (e.g., [38]). During a model run, there were a large number of simulations that clustered close to the optimum for objective performance over a broad range of parameter values (Figure 2). There remains an imperative to further reduce the number of unconstrained model parameters, which is a means of minimizing the impact of equifinality [39,40]. In this work, parameter sensitivity and variability were leveraged as an approach to accomplish the above objective. Only HSP that exhibited significant variability were left unconstrained. The impact of this approach greatly improved performance for both global (1995 to 2015) and final calibration year-by-year (2016 to 2018) series. Therefore, by the application of parameter constraints, objective metrics transcended the limitations imposed within initial model runs by equifinality; or more simply put, by focusing on parameters that really matter, model performance dramatically improved.

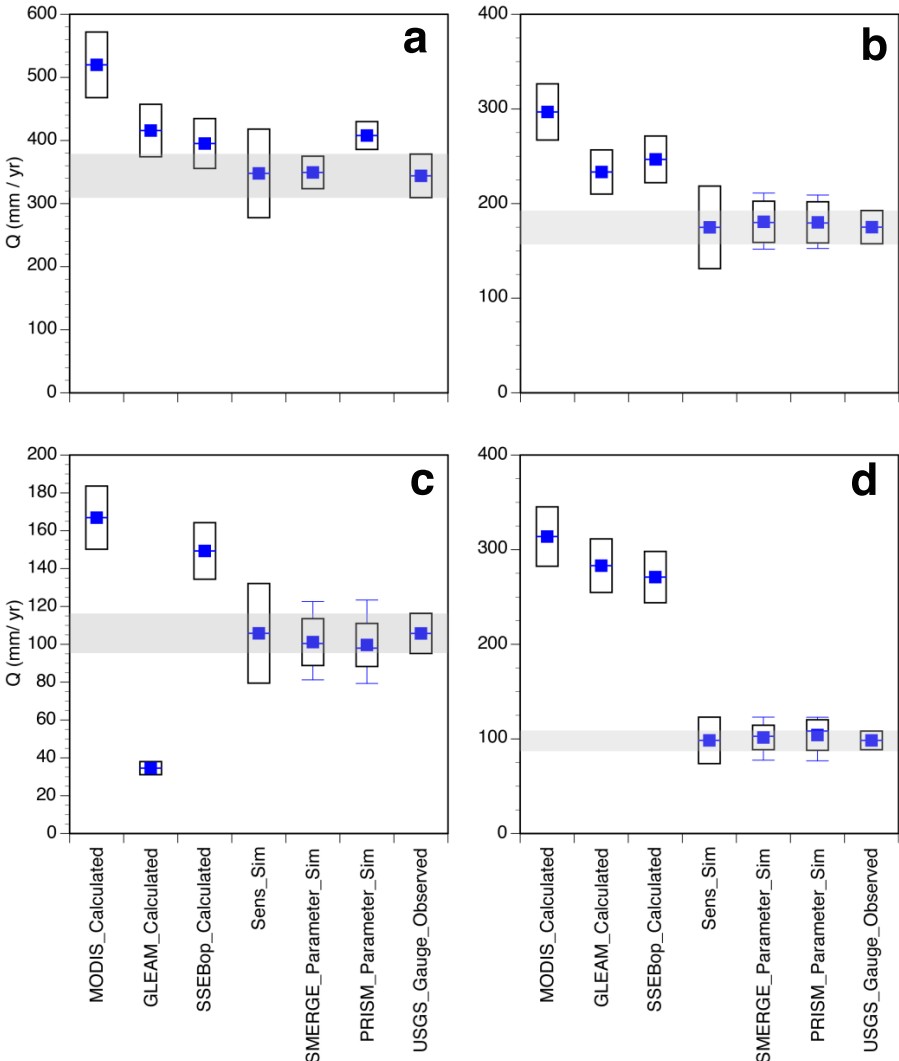

**Figure 5.** Observed USGS, mass balance calculated with the Moderate Resolution Imagining Spectrometer, MODIS16A2v5 (MODIS), Global Land Evaporation: the Amsterdam Model (GLEAM), Simplified Surface Energy Balance (SSEBop), and final calibration year-by-year SWAT simulated (Sens_Q, SMERGE_Parameter, PRISM_Parameter) streamflow (Q). Gray field indicates nominal ±10% error of USGS streamflow observations. (**a**) BC-2017, (**b**) BV-2016, (**c**) BV-2017, and (**d**) BV-2018.

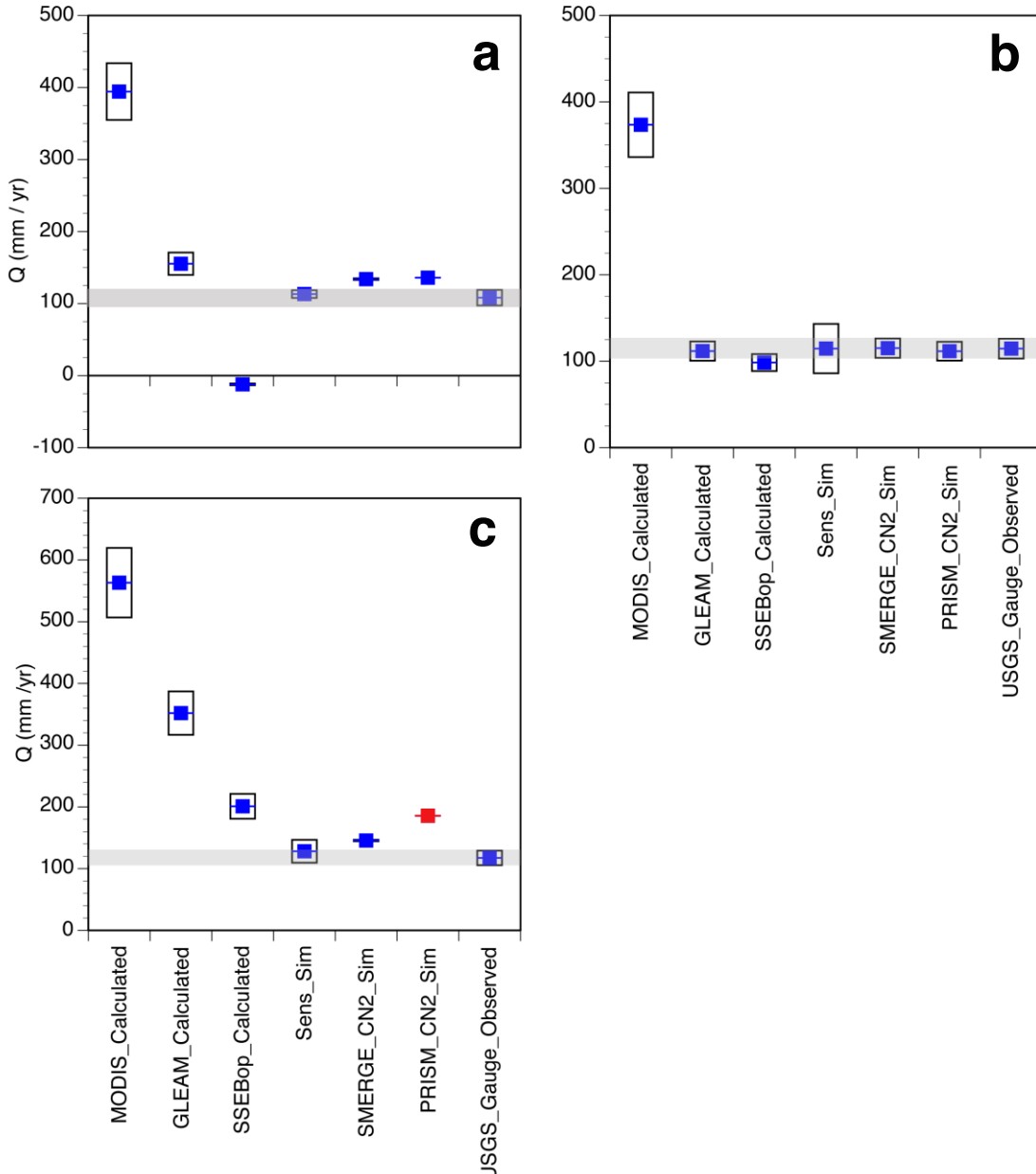

**Figure 6.** Same as Figure 5, except red symbol indicates an unacceptable simulation (RPS < 1.00). (**a**) CH-2016, (**b**) CH-2017, (**c**) CH-2018.

Another interesting result was the slight preference for leveraging RZSM (SMERGE 2.0) over precipitation (PRISM) in improving model performance. This result should not be surprising. While there is no doubt that precipitation is a critical hydrologic variable, no model will yield meaningful results if erroneous precipitation is utilized. Still, it is soil moisture and not precipitation that directly modulates the rainfall–runoff response at a watershed scale. Specifically, antecedent soil moisture strongly governs streamflow response. In all eight examined basins, the Curve Number (CN2) was consistently the most sensitive parameter, emphasizing the importance of soil moisture in controlling streamflow. Indeed, research [41] demonstrates how improved soil moisture accounting by incorporating more realistic Curve Number values can enhance streamflow predictions.

Another approach to minimize equifinality involves the incorporation of remote sensing observations to directly constrain model parameter values [4–13]. Obviously, the utility of this approach relies on the robustness of the remote sensing data applied. As indicated previously, the model structure of SWAT has issues with the direct assimilation of soil moisture data [24]. ET is

another important flux with the water budget at a watershed scale. In the mass balance approach, utilized ET is at least two orders of magnitude greater than ΔS. The fact that PRISM precipitation data was used to drive SWAT simulations and that robust results were obtained supports the general accuracy of this dataset. Therefore, the discrepancies that exist between the calculated and simulated Q values must largely lay with three ET datasets used in this study. MODIS16A2v5 tends to perform less well in the central Great Plains, where there is a more limited vegetation cover [42]. The GLEAM product also had issues in this region. At its core, GLEAM assimilates satellite estimated precipitation that has a strong tendency to overestimate summertime particularly over the Great Plains [43,44]. These errors can generate a strong positive bias in annual ET estimates from this region. Because of these issues, we opted not to utilize data from these ET products to constrain SWAT parameter values.

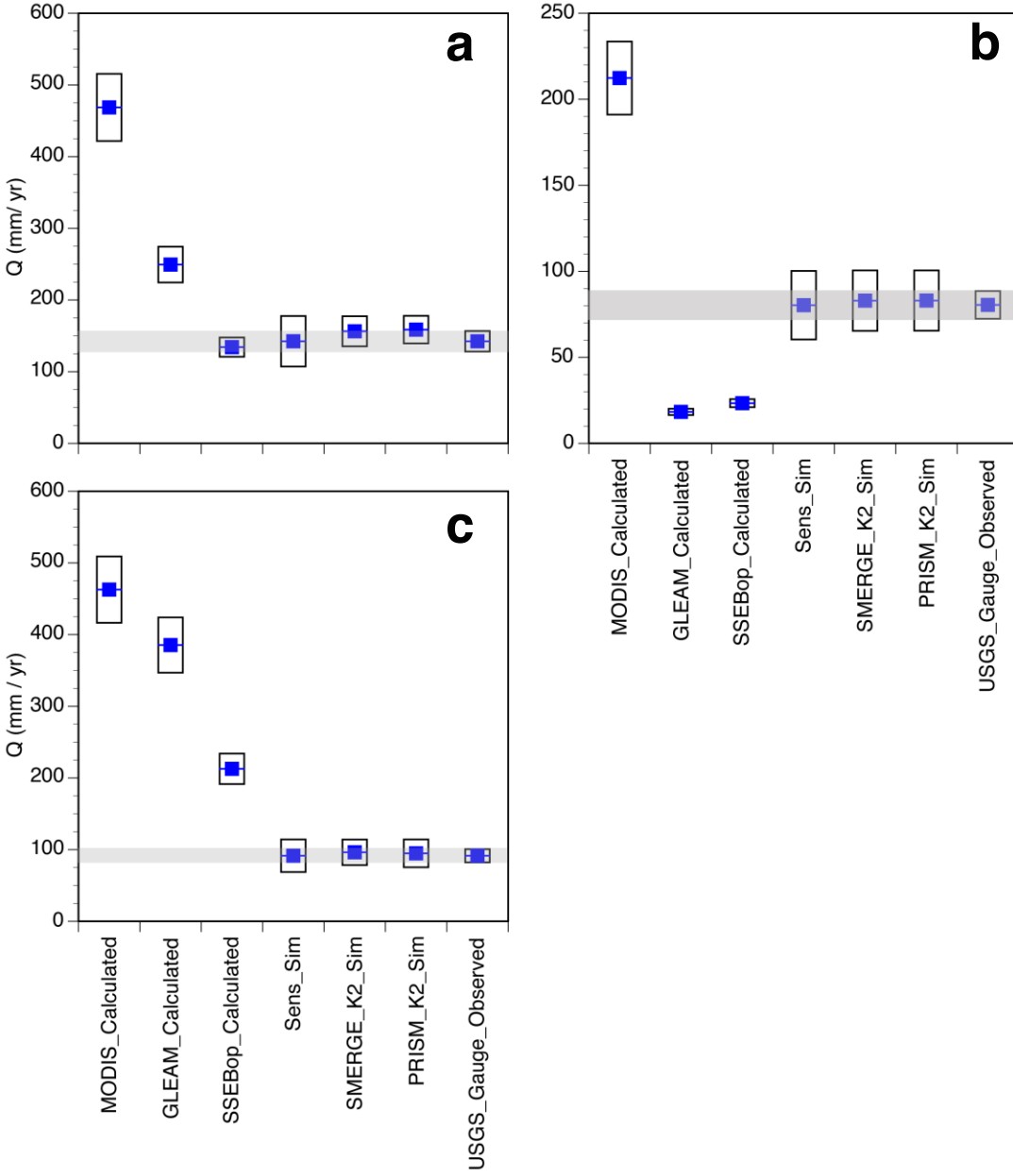

**Figure 7.** Same as Figure 5. (**a**) LA-2016, (**b**) LA-2017, and (**c**) LA-2018.

Future work will further validate the approaches articulated in this study beyond the Great Plains within a broad range of land covers, soil, and climatic regimes. From reference [22], SMERGE 2.0 provided optimal estimates for RZSM in the Southern to Northern Great Plains, the Central Valley of California, and scattered areas in both Southwestern and Southeastern CONUS. In highly forested regions, including much of Northwestern and Eastern CONUS, land surface models like Noah may provide a better estimate of RZSM. In scattered areas from Southern California to Arizona and the corn belt extending from Iowa to Illinois, SMERGE 2.0 also underperforms and land surface model estimates of RZSM are preferable. Interestingly, in no region within CONUS are satellite soil moisture estimates preferred. Either the land surface model or merged land surface model and the satellite soil moisture retrievals yield optimal estimates of RZSM [22].

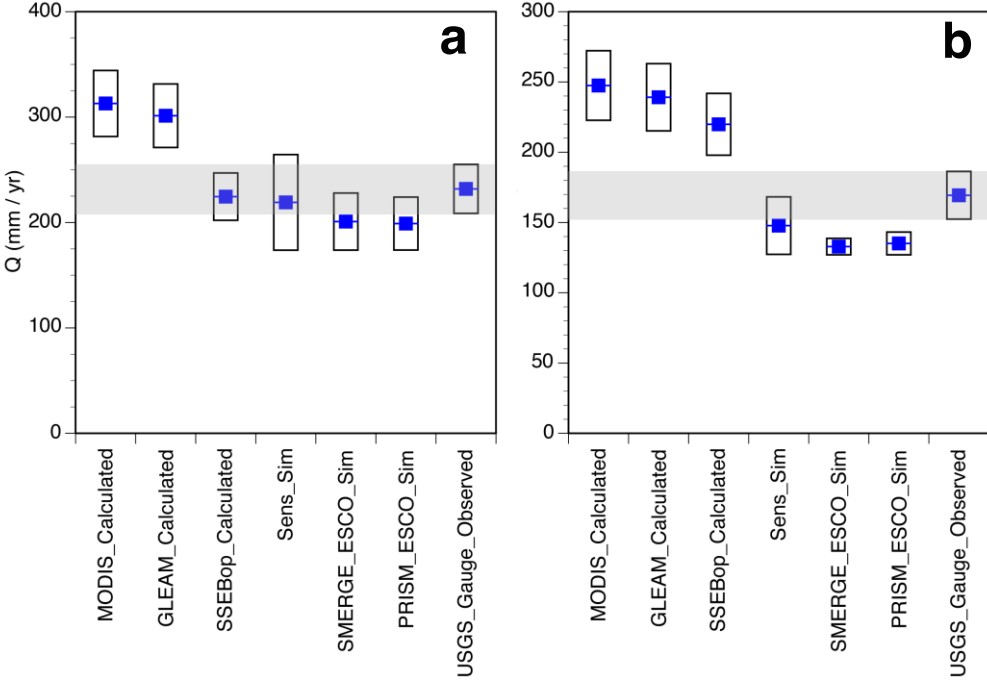

**Figure 8.** Same as Figure 5. (**a**) LN-2016, and (**b**) LN-2017.

## 7. Conclusions

To summarize, the key results are as follows:

(1) The final calibration year-by-year simulation series (2016 to 2018), which was based on executing SWAT on an annual basis, outperformed the global simulations series, in which one objective metric was calculated based on the entire analysis period (1995 to 2015).

(2) For the final calibration year-by-year simulation series, four model runs were executed (Global_Q; Sens_Q; SMERGE_Parameter; PRISM_Parameter). The Global_Q simulation, which was based on parameter values fixed during the global simulation series, underperformed compared with other model runs, in which parameter values were constrained with information derived from individual year-by-year models as well as SMERGE 2.0 RZSM anomaly and PRISM precipitation data.

(3) SMERGE_Parameter simulations had slightly higher RPS values compared with PRISM_Parameter simulations and also better matched with USGS gauge observed Q.

(4) Calculated Q based on a mass balance approach did not consistently match observed Q, unlike SWAT simulated Q. The highest calculated Q was yielded by using the MODIS16A2v5 ET product, followed by GLEAM v.3.3a, and SSEBop v.4, which best matched with observed Q.

The significant implication derived from this work is the demonstration that constraining parameter values can markedly improve SWAT model performance. In addition, that RZSM from SMERGE 2.0 can be leveraged to also greatly improve SWAT model performance. Therefore, this work highlights how diverse remote sensing data can be used to support hydrologic modeling of streamflow at the watershed scale providing more physically realistic results.

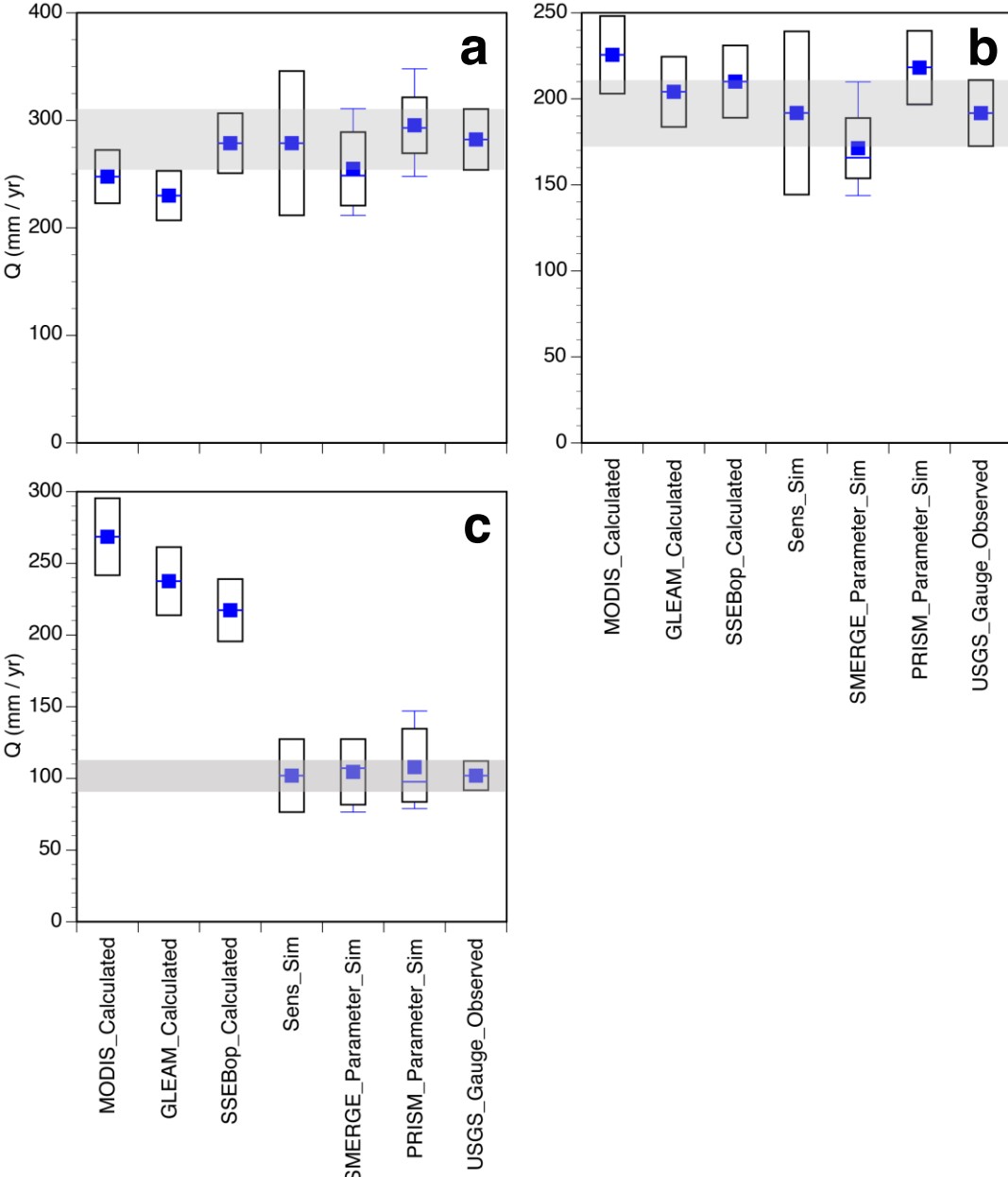

**Figure 9.** Same as Figure 5. (**a**) MC-2016, (**b**) MC-2017, and (**c**) MC-2018.

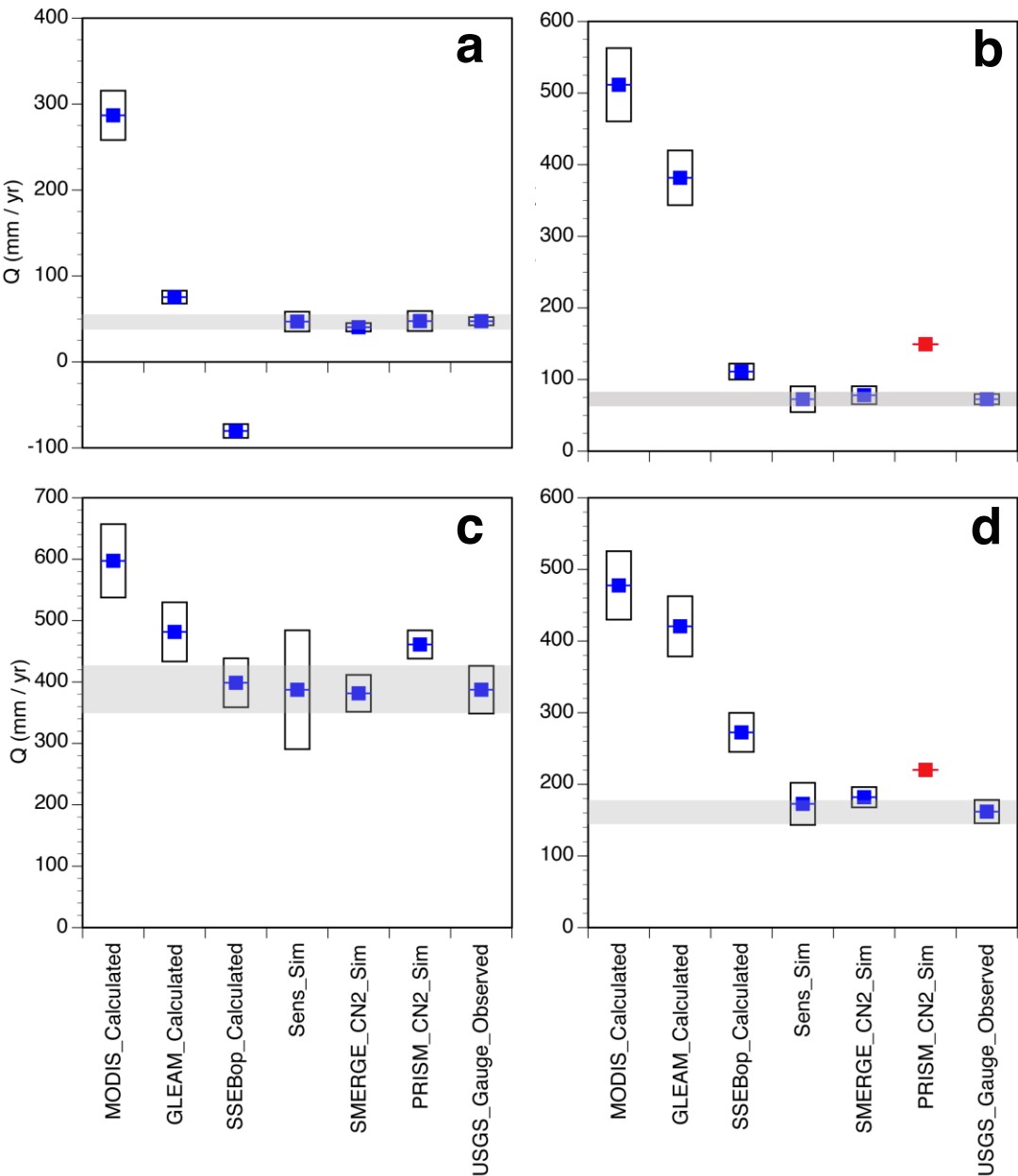

**Figure 10.** Same as Figure 5, except red symbol indicates an unacceptable simulation (RPS < 1.00). (**a**) NI-2016, (**b**) NI-2018, (**c**) WN-2016, and (**d**) WN-2018.

**Author Contributions:** Conceptualization, K.J.T.; methodology, K.J.T.; validation, K.J.T.; formal analysis, K.J.T.; investigation, K.J.T.; resources, K.J.T.; data curation, K.J.T.; writing—original draft preparation, K.J.T.; writing—review and editing, K.J.T., M.E.B.; visualization, K.J.T., M.E.B.; supervision, K.J.T.; project administration, K.J.T.; funding acquisition, K.J.T. All authors have read and agreed to the published version of the manuscript.

**Funding:** NASA Climate Indicator and Data Products for Future National Climate Assessments program through award # NNX16AH30G and NSF Geoscience Equipment (Award Number 1636769).

**Acknowledgments:** The assistance of Arturo Diaz (Texas A&M International University) is greatly appreciated.

**Conflicts of Interest:** The authors declare no conflict of interest. The funders had no role in the design of the study; in the collection, analyses, or interpretation of data; in the writing of the manuscript, or in the decision to publish the results.

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
