# Peer review of "Improving SWAT Model Calibration Using Soil MERGE (SMERGE)"

_water, doi:10.3390/w12072039_

Round 1

Reviewer 1 Report

The intent of this study is good. The paper is fairly well-written. However, the following comments may further enhance the readability of this manuscript:

  1. In the paper title, the last phase “including Soil MERGE (SMERGE)” seems to be out-of-place. Why not use “Improving SWAT model calibration using Soil MERGE (SMERGE)” directly?
  2. What does “RZSM” stand for? All acronym names should be defined (only once) when first appear and be used thereafter. Since this paper utilizes too many of these, why not establish a table to list all these acronym names with their definitions?
  3. The last sentence in the introduction section is redundant. Why not replace it with the real objective of this study?
  4. What is the unit used for “Elevation” in Table 1?
  5. Figure 1 has not been referred to in the text.
  6. Please use “m”, “km” for “meters”, “kilometers”.
  7. How to compute “annual basin-wide averages” mathematically?
  8. Please unify the use of ΔS and DS.
  9. Narrative for Figures 5-10 is extremely lacking. These are the most critical results of this paper.
  10. Discussions on the spatial result comparisons are also extremely lacking. How do the study results vary among all the examined watersheds?
  11. Discussions and conclusions should be separated. In fact, this paper does not include any conclusions just a summary of all the key results. Besides, what are the significant result implications obtained from this study?
  12. A total of 55 reference citations is a bit too many.

Author Response

Reviewer 1

  1. In the paper title, the last phase “including Soil MERGE (SMERGE)” seems to be out-of-place. Why not use “Improving SWAT model calibration using Soil MERGE (SMERGE)” directly?

The title has been modified as requested.

  1. What does “RZSM” stand for? All acronym names should be defined (only once) when first appear and be used thereafter. Since this paper utilizes too many of these, why not establish a table to list all these acronym names with their definitions?

RZSM has been defined the first time it is used in the narrative.

  1. The last sentence in the introduction section is redundant. Why not replace it with the real objective of this study?

Done.

  1. What is the unit used for “Elevation” in Table 1?

The unit “m” was hidden in the table and the table was modified so that it is now shown.

  1. Figure 1 has not been referred to in the text.

Yes, Figure 1 is listed on the beginning of line 69.

  1. Please use “m”, “km” for “meters”, “kilometers”.

Fixed as suggested.

  1. How to compute “annual basin-wide averages” mathematically?

The method used to obtain annual basin-wide averages is now indicated.

  1. Please unify the use of ΔS and DS.

Of course, this has been fixed.

  1. Narrative for Figures 5-10 is extremely lacking. These are the most critical results of this paper.

We added additional descriptions of the results between lines 280 to 309 that should address this concern.

  1. Discussions on the spatial result comparisons are also extremely lacking. How do the study results vary among all the examined watersheds?

We added basin specific results to section 5.1 (lines 240 to 264), which should address this concern.

  1. Discussions and conclusions should be separated. In fact, this paper does not include any conclusions just a summary of all the key results. Besides, what are the significant result implications obtained from this study?

We separated the discussion and conclusions as requested. In addition the significant implications of this study have been better emphasized.

  1. A total of 55 reference citations is a bit too many.

We included as many relevant references as were needed to support our narrative. If the reviewer has some suggestions about what specific references could be removed we will be happy to delete them.

Reviewer 2 Report

This manuscript primarily mainly improving SWAT modeling through identifying highly sensitive parameter values.

The title mentions that including diverse soft data. However, the comparison of various data is not emphasized. A table describing the data sets used should be included. The presentation of results are not that clearly explained and should be improved.

Author Response

Reviewer 2

  1. The title mentions that including diverse soft data. However, the comparison of various data is not emphasized.

The title has been modified as suggested by reviewer 1 (comment 1) to more emphasize SMERGE and less the diverse soft data.

  1. A table describing the data sets used should be included.

We thought about that but already have 10 tables in this paper which is already high side. Also, we are trying to lessen the page count and be a succinct as possible while at the same time convey as much as possible about our work. We believe such as table would largely repeat what is already in the text and therefore opted not to include one in the revised manuscript.

  1. The presentation of results are not that clearly explained and should be improved.

Additions to the text as suggested by reviewer 1 (comments 9 and 10) should address this concern.

Reviewer 3 Report

General Comments:

This study used diverse data to improve SWAT simulation performance using three different kinds of simulation series. The authors used the relative performance scales, which considered both MBE and NSE as the criterion, . The research sounds interesting, and the results showed that the model performance was improved markedly by the constraints of parameter values. However, the scientific writing and organization really hindered my understanding of the research. In addition, here are some major issues that need to be addressed carefully before it can be considered for publication.

Major Comments:

  1. The novelty of the study should be highlighted.
  2. Line 50-51: ‘One of the most important parameters within SWAT is...’ Such kind of statement needs to be supported by relevant literatures. Also, the literature review of this article is not sufficient considering hundreds of publications about SWAT calibration. Below are just a few examples:

Parameter Uncertainty Analysis of the SWAT Model in a Mountain-Loess Transitional Watershed on the Chinese Loess Plateau, Water, 2018

Automating calibration, sensitivity and uncertainty analysis of complex models using the R package Flexible Modeling Environment (FME): SWAT as an example. Environmental Modelling & Software, 2012

  1. Why did this study select these eight regions as the study area? The reason is supposed to be stated in section 2.
  2. The use of punctuation in this full text should be rechecked (e.g., Line 65, Line 95-96, Line 104).
  3. Line 8: The first word ‘A’ should be deleted.
  4. Line 46: The explanation of ‘RZSM’ is supposed to be provided when it first appeared in this manuscript.
  5. Line 151: RPS should be explained, maybe in line 150.
  6. Line 191: Why omit the unacceptable year? How does this method deal with the abnormal years?
  7. Line 200 and 201: Table 4 and 5 have the same title.
  8. Line 244-250: The explanation of the four types should be more clear.
  9. They should use right format of references.

Author Response

Reviewer 3

  1. Rows 61-63: aims of the paper must be clearer written in relation to results at the end;

The end of the introduction has been modified as suggested by reviewer 1 (comment 3) to specify the specific objectives and results presented within the paper.

  1. rows 73-74 and Table 1: not use soils, you mean soil texture; also not dominant land, use Land cover

Yes, there was a problem with the header of the table that has been and soil texture has been added.

  1. Fig. 1: include in the figure river System with subbasins, site with rain gauges and weir for Q

We modified Figure 1 to include subbasins and river drainage with USGS streamflow monitoring sites. Since we used PRISM as a precipitation dataset the location of rain gauged is not relevant to this study.

  1. 4. Datasets are described sufficiently, but a table with averaged input Parameter for each basin is necessary.

After reexamining the methodology carefully, we believe we found the source of confusion in terms of the input parameters the reviewer indicated. For the Base_Q simulation the parameters are left unconstrained as indicated in Table 3. There are no averaged input Parameters values for Base_Q. Parameter values become constrained later during the IT_Q model runs. We have modified the text (lines 171 to 192) to better explain how parameter values become progressively constrained from the Base_Q to iterative model run series.

  1. Ist not clear, how the basin characteristics are different.

As described in Section 2, the eight selected watersheds are fairly similar to each other. Most are loamy soils except the Little Nemaha basin. In terms of land use /cover they are mostly agricultural or rangeland/grassland. The reason these watersheds were selected is that they are largely similar allowing us to validate the methodology in multiple watersheds to confirm that the results were not endemic to a single basin.

  1. Also, in Methods Ist not clear, are the HRU-parameter values averaged for the subbasin or not?

Yes, these HRU-parameter values were averaged at the subbasin level in this study. We have clearly indicated this in the revised paper (line 192).

  1. Include table or figure, how sensitive input parameter as soil moisture or hydraulic conductivity vary in space within a subbasin. The problem with SWAT often is, that spatial heterogenity is leveled out with subbasin averages.

Since we averaged HRU-parameter values at the subbasin level we cannot do the type of analysis suggested. However, to get a sense for how spatial heterogeneity within a basin can impact results we ran SWAT-CUP for the Black Vermillion basin focusing on the CH_K2 parameter by varying it in six subbasins individually. We would have modeled more but wanted to make sure we resubmitted this work in a timely manner. Results from this exercise yields exactly the same optimum CH_K2 value for five out of the six simulations. The sixth model run had the second based result corresponding with the other five model runs. This suggests that the spatial heterogeneity is not that great when looking at different subbasins across this watershed.

  1. Therefore HSP´s can differ on the scale of HRU´s and subbasins. With IT-Q simulations with their range HSP´s are calculated. Ist no surprise, that CN2 is the most sensitive Parameter (Tab. 3). In 5. please insert an example (figure)with the variance of Q (simulated) for the Parameter CN2.

We have included such a figure (Figure 2a) in the revised manuscript and text related to this figure has been added (lines 184 to 185).

  1. Not clear: how antecedent soil moisture in the HRU´s was set for the modeling? Explain more.

On lines 164 to 165 we indicate that a three to four year warming up period was used to initialize the model and establish a baseline for soil moisture conditions in the examined basins.

  1. In the discussion include, how soil moisture data can be used to perform better CN-values for modeling.

This is already discussed in the paper between lines 331 to 339.

Reviewer 4 Report

SWAT-modeling with autocalibration routine (SUFI-2) was used for 8 basins in the Great Plains (USA). The covering period was 1992-2015 for global and yearly Simulation, the calibration period was set to 2016-2018. The worldwide use of SWAT with the HRU-concept take the CN-values in relation to the HRU-characteristics as indirect Parameter for the highly spatial and time related variance of the soil moisture in the upper soil Horizons. Therefore the problem is well written to include with global or national data sets (as SMERGE 2.0) this unsecurity and to improve simulations of Q with the detection of highly sensitive parameter and how data sets from satellite data (e.g. MODIS, GRACE) can be used to improve SWAT simulations.

Specific Points: rows 61-63: aims of the paper must be clearer written in relation to results at the end;

rows 73-74 and Table 1: not use soils, you mean soil texture; also not dominant land, use Land cover

Fig. 1: include in the figure river System with subbasins, site with rain gauges and weir for Q

Datasets are described sufficiently, but a table with averaged input Parameter for each basin is necessary. Ist not clear, how the basin characteristics are different. Also in Methods ist not clear, are the HRU-parameter values averaged for the subbasin or not? Include table or figure, how sensitive input parameter as soil moisture or hydraulic conductivity vary in space within a subbasin. The problem with SWAT often is, that spatial heterogenity is leveled out with subbasin averages.

Therefore HSP´s can differ on the scale of HRU´s and subbasins. With IT-Q simulations with their range HSP´s are calculated. Ist no surprise, that CN2 is the most sensitive Parameter (Tab. 3). In 5. please insert an example (figure) with the variance of Q (simulated) for the Parameter CN2.

Not clear: how antecedent soil moisture in the HRU´s was set for the modeling? Explain more.

In the discussion include, how soil moisture data can be used to perform better better CN-values for modeling.

Results are clear: with IT-Q Simulation results are improved and between Sens-Q, SMERGE and PRISM no great differences exists. Best Simulation results are given with S ens-Q runs. Important result is, that SMERGE2.0 data set give best results.

Author Response

The reviewer 4 is the same as reviewer 3 and I have already addressed these comments.

Round 2

Reviewer 1 Report

The authors have done an excellent job in revising the original manuscript. The revised version is much improved than before. 

All my previous comments have been addressed except at the point on over referencing.

To show you the best way to relief the burden, my suggestion is an example on Line 42. In supporting one statement, you have used 5 references. Why not select just 2-3 important ones. This way, your 55 citations may end up with less than 40.

Author Response

I deleted over ten references. Changes to the manuscript are in blue. These changes should reduce the publication length of the paper by a page.